# An Overview on Truffle Aroma and Main Volatile Compounds

**DOI:** 10.3390/molecules25245948

**Published:** 2020-12-15

**Authors:** Ahmed M. Mustafa, Simone Angeloni, Franks Kamgang Nzekoue, Doaa Abouelenein, Gianni Sagratini, Giovanni Caprioli, Elisabetta Torregiani

**Affiliations:** 1School of Pharmacy, University of Camerino, Via Sant’Agostino 1, 62032 Camerino, Italy; ahmed.mustafa@unicam.it (A.M.M.); simone.angeloni@unicam.it (S.A.); astride.kamgang@unicam.it (F.K.N.); doaa.abouelenein@unicam.it (D.A.); gianni.sagratini@unicam.it (G.S.); giovanni.caprioli@unicam.it (G.C.); 2Department of Pharmacognosy, Faculty of Pharmacy, Zagazig University, Zagazig 44519, Egypt

**Keywords:** truffles, aroma, volatile organic compounds, analytical methods

## Abstract

Truffles are underground edible fungi that grow symbiotically with plant roots. They have been globally considered as one of the most expensive foods because of their rarity, unique aroma, and high nutritional value as antioxidant, anti-inflammatory, antiviral, hepatoprotective, anti-mutagenic, antituberculoid immunomodulatory, antitumor, antimicrobial, and aphrodisiac. The unique flavor and fragrance of truffles is one of the main reasons to get worldwide attraction as a food product. So, the aim of this review was to summarize the relevant literature with particular attention to the active aroma components as well as the various sample preparation and analytical techniques used to identify them. The major analytical methods used for the determination of volatile organic compounds (VOC) in truffles are gas chromatography (GC), proton-transfer-reaction mass spectrometry (PTR-MS), and electronic nose sensing (EN). In addition, factors influencing truffle aroma are also highlighted. For this reason, this review can be considered a good reference for research concerning aroma profiles of different species of truffles to deepen the knowledge about a complex odor of various truffles.

## 1. Introduction

Truffle is a gourmet food used in the haute cuisine world for its valued and pleasant aroma [1]. In scientific jargon truffle is an ascomyceteous fungus belonging to the Tuberaceae family of the Pezizales order. Although some fungi are able to form underground fruiting body, e.g., basidiomycetes, only those of the genus *Tuber* are considered true truffles [2]. The word *truffle* probably derived from the Latin *tubera* the plural of *tuber* that means *lump*, *hump,* or *swelling.* Latin called this fungus *tuber* that probably descends from the word *tumere* to indicate its globoid form [1]. In the *Tuber* genus have been discovered more than 200 species and the most of them remain undescribed mainly because many of the species produce small size fruiting body that are morphologically cryptic and lack of any culinary values [3]. On the other hand, some species are extensively appreciated and hunted in several countries. In Europe occur the most valuable truffles, i.e., *Tuber melanosporum* Vittad. (Périgord black truffle), *Tuber magnatum* Pico (Italian white truffle), *Tuber aestivum* Vittad. (summer or Burgundy truffle), and *Tuber borchii* Vittad. (bianchetto truffle) (Figure 1).

The most important harvested species in China is *Tuber indicum* Cooke and Massee, an Asian black truffle similar to Périgord black truffle but with less flavour. *Tuber oregonense* Trappe, Bonito and P. Rawl., *Tuber gibbosum* Harkness, and *Tuber lyonii* F. K. Butters (pecan truffle) occur in America but they present only restricted market [2,4]. Other species that are popular mainly in the Middle East, Mediterranean basin, and Northern Africa are the so-called desert truffles. These ascomyceteous fungi, belonging mainly to the genera *Terfezia* and *Tirmania*, grow in arid and semiarid areas [5,6].

Truffle depends on a suitable plant host with which generates an obligate symbiosis. In fact, it forms a network composed by branched hyphae connected with the plant roots called ectomycorrhizal. This structure with large surface area allows mutually beneficial exchanges of resources between the plant host and the truffle. Without this association the fungus cannot produce the mature fruiting body and therefore complete its life cycle [7]. These fungi have evolved to attract dispersal agents such as insects and mammals by producing intense aroma [2]. This strong aroma and unique flavour are the reasons why truffles are sought-after in the global market and the inability of the production to satisfy the market demand establishes the high price of this gourmet food. Several factors affect the production and therefore the truffle price such as difficulties in truffle cultivation, season variability, manually harvesting that require trained animals and climate changes [7].

### 1.1. History

In ancient times, truffle was known and consumed by Babylonians, Etruscans, Egyptians, Greeks, and Romans. This fungus was vailed by a great mystery since it was unknown where it comes from [8]. For example, an Aristotle’ disciple, Theophrastus, reported that truffle is born from autumn rains or flash of lightning. One hundred years after Theophrastus, Nicander, a Greek poet, supposed that truffles were silt modified by a source of internal heat. One century and half later, Plutarch believed that they were cooked in the mud by lightning. The truffle mystery was more emphasized by its magical fragrance and presumed aphrodisiac quality. For example, Galen prescribes it to his patients because he believed that it was very nourishing, and it generated a general agitation that conduced to sensual pleasure. During the Middle Ages truffle missed popularity even if some scripts report its usages and how to hunt it. For example, some account books proved that truffle was served during the wedding feast of Charles VI of France and Isabeau of Bavaria in 1385 [9]. At the beginning of 18th century Joseph Geoffroy, pharmacist and botanist, made some observations that have been essential to classify truffle as fungus. His observations were then confirmed by an Italian botanist, Pier Antonio Micheli, who described the truffle spores. Important researchers such as Carlo Vittadini and the Tulasne brothers, considered the founders of the modern mycology, firmly studied truffle during the 19th century [8]. In 1808, Josef Talon, a French farmer, developed the first method for truffle cultivation that consisted of planting seedlings that were collected under the oaks where truffles have been found; only thanks to Auguste Rousseau this idea was disseminated and thousands of hectares of oaks were planted in France [5,10]. The method of Talon has been used for more than 150 years until some French and Italian scientists proposed the technique of nursery seedling inoculation [8].

### 1.2. Cultivation

The first tentative of cultivation, as already mentioned, was performed by Josef Talon during the beginning of 19th century. Using the method of Talon, which consisted of planting acorns under the oaks that produced truffles and transplanting the resulting seedlings in new area, vast plantations were established in France and successively in Europe [11]. The second part of nineteen century is known as the golden age of truffle production also thank to the Talon method while a marked decline, especially in France, occurred in the first half of 20th century. This falling down of production can be probably ascribed to the two World Wars. The inoculation of seedlings in the nursey with various species of *Tuber* together with the usage of spores or segments of infected roots has been a big progress on truffle cultivation [5,12]. Various species such as *T. aestivum, T. borchii* and *T. melanosporum* are cultivated in various countries but *T. melanosporum* still remains the main farmed species in different areas of Europe, and in other non-native places such as Africa, Asia, Australia, and America. Desert truffle such as *Terfezia claveryi* Chatin is cultivated in Spain, Israel, and Abu Dhabi using as plant host *Helianthemum* species [4]. Cultivation of the most expensive truffle, the Italian white truffle *T. magnatum*, is not still as productive as that of other species despite several attempts mainly established in Italy since the early 1980s. This is likely due to some difficulties during plant inoculation with the truffle and contaminations that generally occurred [2]. In fact, some fungi can compete with truffle for capturing space on the host roots in the nursery and later in the field. Therefore, the resulting tree will not produce truffles or solely few fruiting bodies. It is important to point out that to obtain a satisfactory production of truffles is basic to use not only an inoculated plant but a well-infected seedling with only one *Tuber* species [12]. This is the reason why truffle growers have required, since several years, a unique certification protocol in Europe for evaluating and certificating the *Tuber* infected plants by molecular and morphological analysis with the aim to protect the truffle farmer for failures [2]. The selection of host trees and diverse environmental factors play an essential role on achievements in truffle cultivation as well. The most cultivated truffle (*T. melanosporum*) has a high number of host trees belonging to different genera such as *Quercus*, *Corylus*, *Populus*, *Tilia*, *Ostrya*, *Carpinus*, *Cistus*, *Pinus,* and *Cedrus* [13]. This black truffle in Europe grows from near sea level in France up to 1800 m in Spain and there are some discrepancies among truffle growers and experts for the selection of right field altitude of cultivation. Another important parameter seems to be the soil slope since wild truffle generally grows up to 60% of the slope. Probably this condition could facilitate the water drainage. Precipitation and temperature also affect the results on truffle cultivation as well as the characteristic of the soil, e.g., stoniness, acidity and alkalinity, minerals, texture, and structure. An alkaline pH (7.5–8.5) and a granular, well aerated structure with good natural drainage soil are positive conditions for black truffle cultivation [11,14]. Further molecular studies on truffle life cycle and mycorrhizal symbiosis could allow to a more productive cultivation for example by selecting a specific mycelian inoculum adapted for the host plant and climatic and edaphic conditions [2].

### 1.3. Uses

Truffle is widely appreciated and consumed for its unique, valuable, and pleasant aroma besides it is one of the most expensive foodstuffs. In fact, it is known as “underground gold” or “diamond of the kitchen” and it is considered a costly delicacy. Although it can be alone consumed as food it is generally used as spice to enhance the dish savour as aroma flavouring [15,16]. In fact, it is added to different dishes such as meat, pizza, pasta, risotto, and eggs. For example, an Italian special dish is a risotto with white truffle. Truffle can be served raw as thin slices added directly to the dish and or sprinkled as garnish [1]. This delicacy is characterized by short shelf life therefore after harvesting it should be consumed quickly. For this truffle aroma is usually entrapped in oils, butter, and dairy products by using pieces of truffle as food additive. Cacciotta, Pecorino and Sottocenere are some popular Italian truffle speckled cheeses [1,16]. Different “truffle oils” are commercially available in specialty stores, but they are produced by using synthesized chemicals that they occur in truffle aroma fraction. In contrast some olive oil infusions are made with this fresh delicacy and they are usually consumed for dipping bread or drizzling pasta [16]. Truffle can be cooked in various ways as well [1]. For example, in the Arabian Peninsula truffle are eaten in different cooking manners: Fried, boiled with rice, or used as a replacement of meat in cooked vegetable dishes [17]. This fungus is used also for preparing sauces (truffle sauce, ketchup, mustard, sauce with mushrooms), soups, purees, spice (truffle salt), and it can be found preserved in brine and canned [16]. Besides the numerous gastronomic uses, truffle has been employed since antiquity for its presumed multiple biological actions. Desert truffle juice was used since the 10th century in the Arabian Traditional Medicine to cleanse the eyes and to eye inflammations while currently, in ethnopharmacology, it is known for aphrodisiac effects. Moreover, the boiled extract is used for the treatment of trachoma and as anti-inflammatory for eye diseases [17]. Modern scientific literature has evidenced that truffle can be considered a valuable therapeutic agent with antioxidant, anti-inflammatory, antimicrobial, antimutagenic, antitumor, aphrodisiac, and antidepressant activities [1,15,16,17]. This is the reason why it can be employed in pharmaceutical and cosmetic industry [16,18].

### 1.4. Tracing Truffles in the Soil and Interactions with Mammals

The high cost of truffles, arise from their difficult and labour intensive treasure-hunting dedication. In nature, truffles attract mammals ranging from wild pigs or boars to squirrels, which consume the fruiting bodies and contribute to spore dispersal [19,20,21,22]. Truffle hunters have traditionally used pigs and more recently trained dogs to localize the truffles underground. In Italy and France, small groups of truffle hunters scour the woods with dogs and pigs looking for truffles in secret spots. Trained pigs and dogs through their ability to detect and recognize the odorant volatile organic compounds (VOCs), determine the underground locations of truffles. While pigs have the keener nose for truffles, they tend to eat the truffles, so dogs are preferred as they have little appetite for mushrooms. It was reported that a steroidal pheromone, 5α-androstenol, emanating from both black and white truffles with a characteristic musk odor, is responsible for attracting pigs [1,22,23,24]. However, it was later demonstrated that dogs and pigs were attracted not by 5α-androstenol but by dimethyl sulfide (DMS), another compound present in black truffles [19,22]. Because DMS is present in numerous truffle species, it might possibly, along with other unidentified compounds, act as an attractant for mammals in the wild. Therefore, DMS appears to be the key-odor compound for truffle location. Also, the locations of truffles can be detected by observing the hovering of Suillia flies as they lay eggs on the ground above truffles, to provide food for the larvae [1]. Two other sulfurous and three C8 compounds are reported to be attractant for these truffle flies [1,25].

The unique flavor of truffles is one of the main reasons to get worldwide attraction as a food product. Previous studies have focused on screening and identifying the volatile organic compounds (VOCs) and characterized more than 200 VOCs in various truffle species. The major compounds responsible for the aroma in diverse truffle species are well documented in our review which aimed to provide an update on research conducted on the analysis of active aroma components as well as the analytical techniques to identify them. So, this present review focuses on summarizing the up-to-date research on the volatiles of truffles.

## 2. Truffle Aroma Characterization: Black Truffle and White Truffle

Truffles vary in their texture (wrinkled, bruised, smooth, and reticulate) and color (white, brown, and black). White truffles include *Tuber magnatum*, *T. maculatum*, *T. borchii*, *T. dryophilum*, *T. puberulum*, *T. oregonense* (Oregon white truffle), *T. excavatum,* and *T. latisporum*, whereas black truffles include *T. melanosporum* (Perigord Truffle), *T. aestivum* (summer truffle), *T. brumale*, *T. uncinatum* (Burgundy truffle), *T. indicum*, and *T. himalayense* [26,27]. About 30 species of truffle are commercially traded, and because of their rarity and unique aroma, they are one of the most expensive foods in the world [28].

### 2.1. Aroma Profiles of Truffles

The characteristic aroma of truffles vary from mild to intense, and range from earthy, cheese, pungent, garlicky, leathery, vanilla-like, dusty, creamy to gasoline like [27,29]. The volatile organic compounds (VOCs), responsible for the distinctive scent of truffles are a blend of alcohols, ketones, aldehydes, aromatic, and sulfur compounds, albeit only a small fraction of all the VOCs are responsible for what humans perceive as truffle aroma “aroma active compounds”. Many researchers have focused on screening the VOCs, namely the aroma active compounds, and characterized more than 200 VOCs in various truffle species. According to literature, the aromatic profile of a single species typically contains 30–60 volatile constituents [25,30,31,32,33,34].

Aroma profiles of truffles showed a diverse variability and the unique aroma of specific truffle is attributed to its unique components. Specific odorants are common to many truffle species, while others are species-specific or limited to only a few species (Table 1). For example, sulfur-containing volatiles such as dimethyl sulfide and dimethyl disulfide, 2-methylbutanal, 3-methylbutanal, 2-methylbutan-1-ol, 3-methylbutanol, and oct-1-en-3-ol with typical fungal flavor are common to most white and black truffle species [22]. While 2,4-dithiapentane is exclusive to *T. magnatum* [28,35,36]. Also, the thiophene derivatives especially 3-methyl-4,5-dihydrothiophene, and 2-methyl-4, 5-dihydrothiophene are only found in *T. borchii* and they are the important contributors to the human-sensed aroma of this truffle [22,37,38]. In addition, the analysis of aroma active compounds of *T. japonicum* (a novel white-colored truffle native to Japan) showed a high contribution of 1-octen-3-ol and 3-methyl-2,4-dithiapentane to its odor. Although 2,4-dithiapentane is a key odorant of *T. magnatum*, 3-methyl-2,4-dithiapentane was identified as a distinctive sulfur volatile component from *T. japonicum* [39]. The concentration of a single aroma compound can also vary significantly in truffles of the same species and even in truffles collected from the same field [40]. Hence, scientists have documented a significant variability in the concentration of four (2-butanone and 2-butanol), and eight (1-octen-3-one, 1-octen-3-ol, and trans-2-octenal) carbon-containing volatiles in *T. aestivum* fruiting body collected only a few centimeters apart in the same truffle location [28,40]. 

The aroma potency varies according to the truffle type. It has been reported that the black truffles have the highest aromatic rate while summer truffles have the lowest, and white truffles are in between. For this reason black truffles are considered to be the most aromatic of all the truffles [41]. For example, the black truffles *T. melanosporum* is considered to be one of the most aromatic species and it has called a “black diamond of cuisine” due to its potent and complex aroma. This species is very vulnerable to fraud with the Chinese truffle *T. indicum* which looks very similar but it has less intense and complex aroma than *T. melanosporum* so it possesses a much lower market value [42]. Additionally, *T. brumale* is another black truffle with a characteristic musky odor and “earthy” notes. *T. aestivum* (summer truffle) is less aromatic than *T. melanosporum*, although it has a good aroma quality and it is very appreciated by consumers. Lastly, *T. magnatum*, the most expensive truffle on the market, is often considered as the finest species and has a complex aroma reminiscent of garlic and cheese [43].

### 2.2. Factors Influencing Truffle Aroma

Truffles of a given species show significant variability in their aromatic profiles [32]. The differences and variabilities in aroma profiles of truffles arise because of abiotic factors (e.g., soil properties, rainfall and temperature, microclimatology, and mycelia connectivity) and biotic factors (e.g., bacteria, fungi, yeasts, mesofauna, and host plant) that often co-vary in truffle land [44,45]. The fruiting bodies of truffles are produced in the soil in a symbiotic mycorrhizal association with plants and microbes. Bacteria which represent the third component of these associations can produce VOCs that contribute significantly to truffle aroma together with other Tuber-associated microbes (yeast and fungi) [46]. The microbiome on truffles are regarded responsible for the aroma profile [47]. For example, sulfur containing volatiles such as thiophene derivatives, characteristic of *T. borchii* fruiting bodies, originate from the transformation of non-volatile precursors of truffles into volatile compounds via these bacteria [27,48]. There is even an indirect evidence that bacteria might be exclusively responsible for the production of 2,4-dithiapentane in *T. magnatum* [47]. Also, aroma variability in truffles may be attributed to maturation stage (*T. borchii*), environmental or geographical origin (*T. magnatum*) and genetic factors [22,28,49,50,51,52] of truffles. In addition, sample preparation may also influence the aroma profile, for instance, freezing can lead to the de novo formation of volatiles by disrupting tissues and allowing enzymes to work on specific substrates [40]. It has been reported that frozen samples are more abundant in compounds such as diacetyl, 1-octen-3-one, and 1-octen-3-ol [53]. It was found that the effect of storing truffles at 4 °C for up to 6 days was negligible, whereas the freezing process significantly influenced the aroma profile [49]. There is a sensory study revealed a significant reduction in the characteristic fresh aroma of *T. melanosporum* after only 24 h of freezing and methional and some phenols were suggested as markers of freezing time. Interestingly, 1-octen-3-one appeared as a general marker of freezing process [53]. Other factors such as hydration and storage conditions can affect the aroma profile. As truffles are a perishable commodity with a short shelf life (about 1 week), storage temperature and time are major factors influencing truffle aroma [37,54]. Some researchers reported that the number of emitted volatiles strongly increases in truffles stored for a few days at room temperature and they suggested that commensal and spoilage microbes might be directly or indirectly driving the shift in aroma profile observed upon aging, and hence preservation techniques are required [55].

Regarding the industry, another critical factor affecting truffle aroma is the preservation method used. Various processing and preservation techniques are typically used in the industry to protect aroma and help prolong their shelf life after harvest such as freeze-drying [56,57], refrigeration [58], irradiation [59], modified atmosphere packaging (MAP) with microperforated films [60], and Treatment combination (MAP, electron-beam irradiation, refrigeration) [61,62,63]. With the recent progress in food technology, more innovative ways are expected to preserve truffles whilst maintaining their sensory and biochemical properties.

All the aforementioned factors contribute to a certain limit in shaping the characteristic aroma of a specific species. Although the variability in aroma profile, truffles of a given species has common volatiles that can act as fingerprints to identify this species.

### 2.3. Major Truffle Aroma Components

Table 1 presents many aroma compounds identified in different types of black and white truffles and their relative percentages of area reported in literature. Several compounds appear to be species-specific. For instance, 2,4-dithiapentane was detected only in *T. magnatum*, and it is the major contributor to the aroma [49,64,65]. Dimethyl disulfide is mainly present in the aroma of *T. magnatum* and *T. melanosporum*, while 2-methylbutanenitrile, 2-nitropentane, and 2-bromo-2-methylbutane are unique to *T. rufum*, which has a distinctive ethereal, fruity odor [64,65]. 2-Butanone and 2-butanol were detected mainly in *T. aestivum* samples. However, 2-Butanone is considered as a quality marker because all of the samples containing low amounts of 2-butanone had an unpleasant rotten odor, except for samples with a high amount of 1-octen-3-ol, which showed an acceptable aroma [65]. It was reported that the concentrations of C4–VOCs (2-butanone and 2-butanol) or of C8–VOCs (1-octen-3-ol, 1-octen-3-one and 3-octanol) correlate with and are dependent on clones/genets and not on genetic clusters [40]. In an earlier study of *T. aestivum*, the researchers found a strong link between the concentration of C8-containing volatiles and the genotype [49]. 1-(methylthio)Propane is another compound present only in *T. macrosporum* has an alliaceous, creamy green, leek odor. Two compounds typical for *T. macrosporum* are (E)-1-(methylthio)-1-propene, and (Z)-1-(methylthio)-1-propene, which are also present in *T. excavatum*. *T. excavatum* and *T. macrosporum* possess similar aroma compounds just in different ratios. It was reported that 1-methoxy-3-methylbenzene (3-Methylanisole) is typical for *T. mesentericum*, *T. brumale*, *T. indicum,* and *T. excavatum* and has a strong unpleasant spicy odor reminiscent of car paint [64,65]. It is described in literature as having musky-mouldy odor notes, however the odor contribution of anisole cannot be excluded [64]. Anisole was detected mostly in *T. brumale* and *T. melanosporum*. Fresh *T. melanosporum* showed the highest amount of dimethyl sulfide, 2-methylbutan-1-ol, 2-methylpropyl formate, 2-methylbutanal and 3-methylbutanal. 2-Methylbutan-1-ol and 2-methylbutanal are typical for *T. melanosporum* and are responsible for its sulfurous, animal odor [64]. Dimethyl sulfide and 2-methylbutanal mixture was patented to mimic the smell of *T. melanosporum* [47]. Additionally, 57 volatiles were detected in *T. liyuanum* (a novel truffle species), in which aldehydes and aromatics were the main chemical constituents, and 3-octanone, phenylethyl alcohol, isopentane, and methylbutane were more significant in contribution to its aroma [66]. Recently, It was reported that the Chinese white truffle possesses more sulfur volatiles (2.894 μg/g) than the Chinese black truffle (0.040 μg/g) in terms of amounts and number, and sulfur volatiles in white truffle included methional, (methylthio)-cyclohexane, 3-methylthio-1-propanol, 3-(methylthio) propanoic acid and benzothiazole, while black truffle contained dimethyl disulfide and 3-(methylthio) propanoic acid [67]. Generally, the most abundant aroma compounds in most truffle species are dimethyl sulfide, 1-methoxy-3-methylbenzene, 1-octen-3-ol, 3-octanone and 3-methylbutanal. According to the literature, 3-methyl-1-butanol, 1-octen-3-ol, 3-methylbutanal, 3-octanone, hexanal, and acetaldehyde occur in more than 50% of all truffle species [65]. Noteworthy, numerous volatile terpenoids have been detected in fruiting bodies of truffle, especially white truffles, which interestingly present a much larger isoprenoid metabolic diversity than black truffles. Twenty-four isoprenoids (e.g., limonene) have recently been reported in the white truffle *T. magnatum* collected in different Italian regions [68]. Also, the production of 15 isoprenoids at different stages of ascocarps maturation have been reported in another white truffle, *T. borchii* [69]. In contrast, the black truffles *T. melanosporum*, *T. indicum,* and *T. aestivum* seem to contain few isoprenoids [22]. More details about the aroma components detected in truffles are listed in Table 1.

Finally, desert truffles (*Tirmania nivea*, *Terfezia boudieri*, *Terfezia claveryi*, and *Picoa lefebvrei*) are relatives of the highly prized European truffles, but they are less intense in aroma and flavors. The fruit bodies of some desert truffles e.g., *T. nivea* contain the key flavor components such as sulfur volatiles, albeit in much lower levels than in their European truffles (*T. melanosporum*, *T. magnatum*). It has been reported that the main volatiles of *T. boudieri* and *T. nivea* were 1-octen-3-ol and hexanal; however, volatiles of the latter species further included 2-methylbutanal, 3-methylbutanal, benzaldehyde and benzenacetaldehyde, methional and dimethyl disulfide. The weak aromatic truffle, *P. lefebvrei*, contained the main volatile, 1-octen-3-ol in low levels. It was mentioned that the total volatile levels in *T. nivea* were about 2-fold higher than in *T. boudieri*, and *T. boudieri* has about 10-fold higher volatile levels than in *P. lefebveri* [70]. 

**Table 1 molecules-25-05948-t001:** Volatile organic components (VOCs) identified in different types of black and white truffles and their relative percentages of area according to literature.

No	Compound Name and Class	Odor Description	Content [%]										References
			Black Truffles							White Truffles			
			*T. mel*	*T. aest*	*T. ind*	*T. mac*	*T. mes*	*T. ruf*	*T. bru*	*T. mag*	*T. exc*	*T. bor*	
1	**Sulfur compounds** Dimethyl sulfide	Rotten, cabbage	0.52, 57.0	1.04, 0.26	8.3					2.95, 4.44			[31,42,52,53,71]
2	bis(Methylthio)methane (=2,4-Dithiapentane)	sulfuric, garlic								67.33, 83.71, 67.8			[52,65,71]
3	Methyl (methylthio)methyl disulfide									0.05, 0.06			[52,71]
4	Dimethyl disulfide	Cabbage, onion	0.04, 0.17	0.06						0.15			[31,53,71]
5	Dimethyl trisulfide	Rotten food	0.07, 0.01	0.18						0.02			[31,53,71]
6	Sulfinylbismethane		0.04	0.05									[31]
7	Methanethiol	Cabbage, vegetal	0.17							0.010			[53,71]
8	tris(Methylthio)methane									0.29			[71]
9	3-Methyl-4,5-dihydrothiophene	Onion, savory, roast, truffle, garlic, butter										4.87	[48]
10	2-Methyl-4,5-dihydrothiophene	Aged cheese, rubber										0.34	[48]
11	DMTS = (methyltrisulfanyl)methane	Truffle, onion, garlic										0.04	[48]
12	Methional	Boiled potatoes	0.12										[53]
13	Dimethylsulphoxide	Cheesy, garlic, mushroom	0.33										[53]
14	(*E*)-1-Methylsulfanylprop-1-ene	Acrid strong garlic-like		0.2		27.0	0.3				29.6		[65]
15	1-Methylsulfanylpropane	Alliaceous creamy leek		0.1		22.1					0.4		[65]
16	Methylsulfanylmethane		11.7	18.9	4.3	1.3	1.2	8.2	4.5	14.6	4.9		[65]
17	(*Z*)-1-Methylsulfanylprop-1-ene	Acrid strong garlic-like				8.0	0.2				6.7		[65]
18	(Methyldisulfanyl)methane		0.2			0.2			0.1	2.8	0.1		[65]
19	4-Mercapto-4-methyl-2-pentanone		0.06										[31]
20	**Alcohols** 1-Octen-3-ol	Mushroom, earthy	4.04, 0.01,2.0	35.2, 1.8, 0.9	37.1, 0.2	4.7	9.6	3.1	11.2	0.09	2.4	2.51	[31,42,48,52,53,65]
21	2,6-Dimethyl-2-octanol									0.26			[52]
22	*Z*-5-Octen-1-ol				2.8								[42]
23	3-Octanol	Earthy, mushroom, herbal	0.05	4.03, 0.11	2.1								[31,42,52]
24	1-Octanol	Waxy, green, citrus		0.41									[52]
25	2-Butanol		0.03	0.23, 6.3				0.3	0.1				[31,65]
26	6-Dodecanol		1.48										[31]
27	Octa-1,5-dien-3-ol		0.19	0.32									[31]
28	1-Hexanol	Alcoholic, pungent, green	0.07	0.51									[31]
29	2-Ethyl-1-hexanol		0.02	1.30									[31]
30	Ethanol									0.06			[71]
31	Isobutyl alcohol									0.22			[71]
32	2-Methylbutan-1-ol	Malty	4.94	4.7, 1.4, 3.7		0.6		3.2	0.4	3.70			[31,52,65,71]
33	3-Methyl-1-butanol (Isoamyl alcohol)	Winey, onion, cheese	41.9		13.9, 6.5			5.4		0.2	1.8	0.29	[42,48,53,65]
34	2-Methyl-1-propanol	Winey	0.31	0.59									[31]
35	2,5-Dimethyl-3,4-hexanediol				0.7								[42]
36	**Acids** Nonanoic acid			0.05									[52]
37	Acetic acid			3.17									[31]
38	2-Propenoic acid			1.99									[31]
39	2-Methylhexanoic acid		0.08	0.09									[31]
40	**Esters** Ethyl acetate		0.04	0.47									[31]
41	Ethyl-3-methylbutanoate	Fruity, anise	0.09	0.21									[31,35]
42	Butyl-2-methylbutanoate		0.04										[31]
43	2-Methylpropyl 2-methylbutanoate		0.29										[31]
44	2-Methylpropyl-3-methylbutanoate		0.03										[31]
45	3-Methylbutyl-2-methylpropanoate		0.27										[31]
46	2-Methylbutyl-2-methylbutanoate		2.49										[31]
47	Pentyl-3-methylbutanoate		0.29										[31]
48	*sec*-Butylformate				1.80								[42]
49	Ethyl 3-methylbutyrate	Fruit, anise	0.03										[35,42,53]
50	Butan-2-yl formate		15.70		11.40	0.20		2.20	7.00				[65]
51	2-Methylpropyl formate		1.00					1.40	0.90				[65]
52	**Aldehydes** 2-Octenal	Green, citrus, fatty	1.87	0.27, 8.27						0.13, 0.05			[31,35,52,71]
53	Acetaldehyde		0.07	0.28						0.12			[31,71]
54	Nonanal	Waxy, aldehydic, fatty	0.12	1.16						0.07, 0.03			[31,35,52,71]
55	Propanal	Vegetable, green	0.24							0.89			[31,71]
56	2-Methyl-butanal	Cocoa, almond-like	19.13, 8.4	0.64, 0.1	2.0			0.4	0.8	0.57, 0.1			[31,42,65,71]
57	3-Methyl-butanal	Green, nutty, cocoa	38.31, 6.5	7.56, 1.2	3.2, 2.8			1.6	0.6	2.15, 1.5	0.8	0.37	[31,42,48,65,71]
58	2-Butenal	Green, vegetable	3.45	7.90									[31,35]
59	Hexanal	Leafy, fruity, sweaty	5.93	17.63						0.074			[31,35,71]
60	2-Methyl-2-butenal	Fruity, almond, nutty	1.22	1.35									[31,35]
61	Heptanal		0.16	5.36						0.05			[31,71]
62	Octanal	Waxy, orange, peel		0.86	0.70					0.03			[31,42,71]
63	2-Heptenal	Fresh, fatty, green	0.35	2.16									[31,35]
64	Decanal			1.15						0.06			[31,71]
65	2,4-Nonadienal		0.11	0.24									[31]
66	Dodecanal			1.71									[31]
67	2,4-Decadienal		0.14										[31]
68	**Ketones** 3-Hydroxy-2-butanone	Butter, cheese, caramel								7.58		0.001	[48,52]
69	2-Octanone	Earthy, herbal								0.07			[35,52]
70	2-Nonanone									0.37, 0.01			[52,71]
71	2-Decanone									0.32			[52]
72	Undecanone		0.05	0.18, 0.62						0.77			[52]
73	2-Butanone		1.35, 1.3	2.2, 38.2, 53.3	1.6, 0.3	0.2	0.3	0.8	0.6	0.56	0.4		[42,52,65,71]
74	3-Octanone	Herbal, lavender, mushroom	0.20, 4.2	1.7, 0.38, 2.6	4.6, 0.2	7.3	4.2	5.3	8.8	0.05, 0.3	2.0	8.29	[31,42,48,52,65,71]
75	2-Propanone		0.09	0.57									[31]
76	3-Penten-2-one		0.44										[31]
77	5-Methyl-2-heptanone		0.21										[31]
78	2,3-Octanedione		0.06	0.06									[31]
79	3-Hydroxy-2-butanone		0.06										[31]
80	3-Octen-2-one		0.37							0.015			[31,71]
81	1-Octen-3-one	Mushroom, earthy, musty	0.03		0.70								[35,42,53]
82	**Aromatic compounds** Benzeneacetaldehyde	Honey, sweet, floral	0.27	0.19, 1.85	1.10					0.11, 0.18			[31,35,42,52,71]
83	1-Methoxy-3-methylbenzene	Narcissus	2.29, 4.2	3.7	44.50	10.60	69.5	17.4	28.3	9.05, 0.3	37.9		[31,52,65]
84	Bis(2-Methylpropyl) ester 1,2-Benzenedicarboxylic acid			1.887						0.81			[52]
85	Dibutyl phthalate			0.29						0.25			[52]
86	Ethylbenzene			0.66									[52]
87	1,4-Dimethylbenzene			7.19, 0.23									[31,52]
88	1-Ethyl-2-methyl-benzene			0.24									[52]
89	Phenylethyl alcohol	Floral, yeast, rose	0.19	0.70, 3.82								0.002	[31,48,52]
90	2,4-Dimethylphenol			0.93									[52]
91	1-Methyl-3-(1-methylethyl)benzene			0.68									[52]
92	Acetophenone			1.07									[52]
93	Ethylbenzene			0.33									[31]
94	1,3-Dimethylbenzene			0.11									[31]
95	1,2,4-Trimethylbenzene		0.07	0.20									[31]
96	Methoxybenzene		0.83										[31]
97	2-(1-Methylethyl)phenol		0.19										[31]
98	Benzaldehyde	Sweet, bitter, almond	1.44	6.94						0.06			[31,35,71]
99	4-Hydroxycroman		0.38										[31]
100	1,2-Dimethoxybenzene		2.66										[31]
101	1,3-Dimethoxybenzene		0.15										[31]
102	Naphtalene		0.04	0.71									[31]
103	2,5-Dimethoxytoluene		0.17										[31]
104	3,4-Dimethoxytoluene		0.12										[31]
105	1-Methoxy-4-(1-propenyl)-benzene		0.04										[31]
106	2-Methoxy-4-ethyl-6-methylphenol		0.03										[31]
107	2,6-Bis(1,1-Dimethylethyl)-4-methylphenol		0.59	3.74									[31]
108	*p*-Cresol	Phenolic/leather	0.03, 0.02										[31,35,53]
109	*α*-Ethylidene-phenylacetaldehyde		0.18	0.60									[31]
110	Phenol			1.37									[31]
111	1,2-Dimethoxy-4-(2-propenyl)benzene		0.16										[31]
112	1-Methyl-4-(phenylmethyl)benzene		0.07										[31]
113	*p*-Methyl anisole									0.15			[71]
114	*m*-Anisole		7.1		0.8, 1.4	0.80		0.20	3.40		0.70		[42,65]
115	3-Ethyl-5-methylphenol	Leather	0.02										[35,42,53]
116	1,4-Dimethoxybenzene	Sweet green hay newly mown hay	0.10	0.10		2.70	0.90	0.70	12.10	0.10	2.80		[65]
117	1,4-Dimethoxy-2-methylbenzene		0.10		1.20	1.60	7.80	0.10	1.40		0.10		[65]
118	**Furans and Furanones** 2-Acetyl-5-methylfuran									0.57, 0.27			[52,71]
119	2-Pentylfuran	Fruity, green, earthy	0.07	0.35						0.05			[31,35,71]
120	2,3-Dihydro-4-methylfuran		0.08										[31]
121	2-Furancarboxaldehyde			0.18									[31]
122	3-Butyldihydro-2(3H)-furanone									0.14			[52]
123	2(3H)-Dihydrofuranone		0.25	4.49									[31]
124	**Alkanes and Alkenes** Decane									0.21			[52]
125	Dodecane									0.08			[52]
126	2-Methylbutane				3.30								[42]
127	Octylcyclopropane				2.20								[42]
128	Undec-1-ene			0.10		0.40			2.90	1.80	1.80		[65]
129	**Terpenes** Camphor									0.07			[52]
130	Limonene	Citrus, orange, fresh, sweet								0.16			[35,71]
131	*p*-Cymene									0.008			[71]
132	2-Methylisoborneol	Mould, earth	0.02										[35,53]
133	**Others** 3-Methyl-1H-pirazol		0.07										[31]
134	3-Ethyl-4,5-dihydro-1H-pyrazole		0.13										[31]
135	2-Methylbutanenitrile					0.10		21.90					[65]
136	2-Nitropentane							3.90					[65]
137	2-Bromo-2-methylbutane							2.30					[65]

*T. mel* (*T. melanosporum*); *T. aest* (*T. aestivum*); *T. ind* (*T. indicum*); *T. mac (**T. macrosporum*); *T. mes* (*T. mesentricum*); *T. ruf* (*T. rufum*); *T. bru* (*T. brumale*); *T. mag* (*T. magnatum*), *T. exc* (*T. excavatum*); *T. bor* (*T. borchii*).

## 3. Analytical Methods for the Determination of VOC in Truffles

The volatiles fingerprints of the captivating aromas of truffles have been assessed in various studies using different analytical techniques. Table 2, reports the major analytical methods used for the determination of VOC in truffles. These methods are based principally on three analytical techniques: Gas chromatography (GC), proton-transfer-reaction mass spectrometry (PTR-MS), and electronic nose sensing (EN) [41].

### 3.1. Gas Chromatography Methods for the Analysis of VOC in Truffles

GC-based methods are the most used analytical methods for the analysis of aroma-active compounds from truffles. The GC analysis of truffles volatiles requires a sample preparation step for VOC extraction followed by their separation, detection, and quantification through GC coupled specific detector systems [81].

#### 3.1.1. Sample Preparation Techniques

The collection of VOC is an essential stage for the GC analysis of truffles. Various sampling techniques can be used for the extraction of truffles aroma compounds.

##### Static Headspace Extraction for VOC Extraction in Truffles

Static Headspace extraction (HS) is a simple extraction technique coupled with GC to analyze the VOC of truffle samples. In static HS sampling, some grams (e.g., 4 g) of sliced truffles are placed in a closed vial (10–20 mL) at a specific temperature for a fixed time [82]. The principle of this sampling technique is to reach an equilibrium state in which volatiles concentrations are equilibrated between truffle samples and the gaseous phase above. At this equilibrium stage, a gas syringe is introduced in the vial HS to collect an aliquot of the gaseous phase containing the volatiles of the truffle sample. The extracted VOC are then transferred into the GC injector via the HS syringe [83]. The static HS extraction efficiency depends on the temperature, the time to reach equilibrium, the level of truffles inside the vial, and the volatility of the individual VOC [84]. This sampling technique is simple, green (no solvent or reagent used), and easily automated with a low risk of artifacts. Moreover, volatile compounds in the HS are collected in a non-discriminant way, allowing thus, to determine a wide volatile profile of truffles [84]. However, static HS efficacy is limited to highly volatile compounds and shows a lower sensitivity due to the low concentration of the VOC collected. Therefore, VOC at low concentrations could not be detected and must require extraction techniques allowing sample concentration [85].

##### Purge-and-Trap VOC Collection Technique

Purge-and-trap (PT) sampling is a type of dynamic HS extraction technique in which an inert gas is purged to the sample to extract VOC, which are then concentrated into an adsorbent trap. Volatiles are desorbed by heating the trap and are injected into the GC [86]. For truffles analyses, pieces of truffle samples placed in a sealed vial are purged by a stream of helium or nitrogen for a fixed time at a specific flow and temperature. Volatiles released in the HS are carried into a trapping system kept at low temperatures (e.g., 0 °C, −100 °C) during purging. Different types of adsorbent traps are used such as cryogenic trapping systems and styrene-divinylbenzene resins (e.g., LiChrolut ethyl vinyl benzene-divinylbenzene trap) [42,79]. The purges-and-trap sampling in truffle analysis presents the advantage to be more sensitive than static HS through the concentration of analytes in the trap. Moreover, the purging gas accelerates the mass transfer and thus the extraction of high molecular weight compounds with low volatility [87]. Pacioni et al. (2014) reported that the use of PT sampling in truffles analysis allows the acquisition of an aroma profile similar to natural olfactory perception since it was performed at room temperature [79]. However, this extraction technique is not efficient for the extraction of highly volatile compounds, which are not concentrated in the resins trapping system. Furthermore, it involves more time, a high sample size (7–21 g) and is without automation [88].

##### HS-Solid Phase Microextraction (HS-SPME) Technique

HS-SPME is the most used sampling technique coupled to GC for the extraction of VOC in truffles. For HS-SPME, truffle samples are either cut into thin slices [89] or frozen with liquid nitrogen and directly grounded to obtain a fine powder [72]. Small quantities of the obtained samples (150 mg–1.5 g) are placed in a screw cap closed vial (10–20 mL) and maintained under agitation at a fixed temperature (25–60 °C) for a specific time of equilibrium (5–20 min) before extraction. NaCl solution can also be added to the grounded sample to prevent enzymatic reactions and improve the release of VOC in the HS [70]. After equilibrium, volatiles extraction is performed exposing an SPME fiber in the HS of the sample for a specific time (10–30 min) [28,49,52]. Following VOC extraction, the SPME fiber is desorbed in the GC injection port. The profile of the VOC extracted depends on the characteristics of the SPME fiber coating. Different types of SPME fibers have been used for truffles extraction, however, 50/30 μm divinylbenzene/carboxen/polydimethylsiloxane (DVB/Car/PDMS) is the most used SPME fiber allowing the extraction of a broad range of VOC polarities thanks to its triple-phase coating [90]. Beyond the fiber type, HS-SPME can be influenced by the time and temperature of equilibrium, the time of fiber exposition, the distance between the fiber and the sample, the size of truffle samples, and the dimension of the HS vial [91].

The common use of HS-SPME for truffle extraction is explained by the high sensitivity of this extraction technique, which allows the enrichment of VOC on the fiber coating in fast extraction time. Moreover, compared to other HS extraction techniques (static HS and PT), HS-SPME, is faster, requires lower sample sizes, can be easily automated, and drastically reduces the level of air and moisture introduced into GC during sample injection [92]. Moreover, due to the specific characteristics of each fiber, HS-SPME is highly selective for the VOC with high affinity with the fiber coating. This selectivity is ideal for targeted analysis but is a limitation in the case of volatiles fingerprinting, causing a partial extraction of the VOC profile of truffle samples [93].

##### Solvent-Assisted Flavor Evaporation (SAFE) Technique

SAFE is an extraction technique based on the distillation of liquid extracts under a high vacuum to isolate odor-active organic compounds. The distillate is then dried before concentration under nitrogen (until 0.5–1 mL), and an aliquot of the extract (1 μL) can be injected in GC for analysis [94]. Zhang et al. (2016) proposed the combination of direct solvent extraction and SAFE (DSE-SAFE) to assess the aroma profile of different Chinese truffles [67]. DSE consisted to extract freshly cut truffle cubes with diethyl ether. The extracts obtained were then subjected to SAFE. The SAFE technique offers the advantage to extract volatile and non-volatile compounds, giving thus a wider profile of flavor components from small polar compounds to larger non-polar compounds [95]. However, the extensive sample handling during SAFE, reduce the reproducibility of this extraction technique. Moreover, the lack of sampling automation and the long sample preparation time required, are the main reasons explaining the reduced use of DSE-SAFE in the extraction of aroma compounds in truffles [96].

#### 3.1.2. Gas Chromatography (GC) Separation

After sampling, the qualitative and quantitative analysis of the aroma compounds from truffles can be performed in GC. From the complex mixture extracted, the organic compounds are separated inside the GC column according to their boiling point and polarity differences. The choice of the column is essential for analytes separation, and different analytical results can be obtained from the variation of the stationary phase of a GC column [81]. For truffles analysis, the most used GC columns are HP-5MS, a low-polarity GC column made of polydimethylsiloxane (PDMS), and DB-Wax, a highly polar column made of polyethylene glycol [41]. In addition to the GC column choice, VOC separation is also influenced by oven temperature ramps. After a specific retention time (RT), each compound elutes from the column and is then revealed by a detector. Newer GC methods involve comprehensive two-dimensional gas chromatography (GC × GC) to improve the separation of truffles odor-active compounds and prevent VOC coelutions [97]. Indeed, GC × GC uses the combination of 2 GC columns with different polarity to improve the chromatographic resolution and thus, broader the spectrum of detected and separated VOC in truffles [98]. For example, Costa et al. (2015) observed that GC × GC allows the detection of VOC from white truffles such as acetaldehyde, β-pinene, and 6-methyl-5-hepten-2-one, which were not separated and detected in normal monodimensional polar or non-polar GC column [71].

#### 3.1.3. GC Detecting System in Truffles Analysis

Eluting at different retention times (RT), each compound is then revealed by a detector [97]. Various detector systems have been used for GC analysis of truffles.

##### GC-Flame Ionization Detector (GC-FID)

FID is a generally used detector in the GC analysis of volatile compounds. Indeed, based on the combustion of analytes, FID produces from each burnt compound, many ions almost proportional to the number of carbon atoms burnt [98]. Thus, this detector is commonly preferred for the semi-quantitative and quantitative analysis of VOC from truffles. Splivallo et al. (2012) performed the analysis of eight-carbon-containing volatiles (C8-VOC) in truffles through a GC-FID system using an internal standard for their quantification [49]. Similarly, Costa et al. (2015) determined the relative abundance of each VOC in the volatile profile of white truffle [71]. However, FID does not provide any structural information on the revealed compounds. Therefore, the combination of FID with other detector systems, which affords qualitative information on detected VOC, is suitable for the identification and quantification of aroma compounds from truffle samples.

##### GC-Mass Spectrometry (GC-MS)

MS is the most used detecting system in the GC analysis of truffles providing the MS spectrum of each eluted VOC. Using MS, the identification of aroma compounds is confirmed by comparing their retention indices (RI) and MS spectrum with mass spectral data obtained from reference databases (NIST, WILEY, ADAMS, FFNSC), literature, and pure analytical standards. The RI is determined using a series of straight-chain alkanes analyzed in the same conditions of truffle samples [70]. Quadrupole mass spectrometer (qMS) [49] and Ion trap mass spectrometer (IT-MS) [72], are the principal mass analyzers used for truffles analysis. Newer generations of MS analyzers such as High-Resolution Time-of-Flight Mass Spectrometry (HR-TOF/MS) allowed the identification of a broader range of VOC from truffles [27]. This analyzer offers the advantage of giving higher mass accuracy and higher mass scan efficiency than qMS and IT-MS [81].

##### Gas Chromatography–Olfactometry (GC–O)

Structural information and quantification of VOC from truffles are provided by GC-FID and GC-MS. However, these detecting systems do not give any information on the odor active volatile compounds impacting the aroma perception of truffles [73]. For this purpose, GC-O can be coupled to FID and/or MS. In GC-O, each eluted VOC is sniffed and detected by a trained human assessor to describe its perceived odor, its intensity, and the duration of the odor activity [74]. Various studies reported the use of GC-O to assess the aromatic profile of truffles [35,75]. Using GC-O, Feng et al. (2019) determined that dimethyl sulfide, 3-methylbutanal, benzeneacetaldehyde, and eight-carbon-containing volatiles (3-octanone, octanal, 1-octen-3-one, and 1-octen-3-ol) are the keys aroma compounds of three Chinese truffles varieties [35].

### 3.2. Analysis of VOC from Truffles Using Proton-Transfer-Reaction Mass Spectrometry (PTR-MS)

PTR-MS is a valuable tool in volatile metabolomics, allowing the identification and quantification of VOC from various foods. In PTR, gas-phase analytes are transported into the proton transfer reaction cell, where analytes protonation occurs from the reaction between VOC and H_3_O^+^. The protonated compounds are then transferred into the mass analyzer for detection without subsequent fragmentation [99]. This soft ionization method allows the direct and absolute quantification of analytes even without the use of calibration standards. PTR-MS application is growing in the analysis of truffle aroma compounds to characterize the VOC fingerprint of truffles or the differentiation of truffle cultivars [76,77]. This analytical technique is an effective alternative to GC-based methods with various advantages. Indeed, PTR-MS being a direct injection mass spectrometry technique (DI-MS), does not require lengthy sample preparation and consents to perform a faster, direct, and real-time (on-line) analysis of VOC. Moreover, PTR-MS shows higher sensitivity than GC-based methods allowing the detection of VOC at pptv concentration levels (parts per trillion by volume) [81]. Besides, the use of high-resolution mass analyzers such as TOF improves PTR-MS technique and provides higher mass resolution. This higher sensitivity allowed Vita el al. (2015) to identify 22 new compounds in the volatile profile of truffles [46]. However, PTR-MS presents the disadvantage to have limited resolving power, being not able to separate isomers, rendering thus ambiguous the identification of some VOC. Therefore, for a more complete description of VOC from truffles, PTR-MS can be combined with GC-based methods [78].

### 3.3. Electronic Nose (EN) Sensing in the VOC Analysis of Truffles

EN is a commonly used analytical technique for the analysis of the truffle aroma. EN is composed of 3 major elements: A sample delivery system, a detecting system, and a data processing system [100]. The delivery system is based on GC sampling techniques such as SHS, DHS, PT, and SPME, which extracts the VOC in the HS of truffle samples and transfers them to a detecting system. The detecting system is made of sensors, whose reaction with VOC, causes electronic responses, which are converted into digital values. The data obtained will then be processed on statistical models [101]. For truffle aroma compounds, the most utilized sensor types in EN analysis are metal-oxide sensors (MOS) [54,79] and quartz microbalance (QMB) sensors [51,71]. The application of EN in truffles includes the quality assessment, discrimination of truffle types, and truffle spoilage control. Pennazza et al. (2013) performed EN analysis using a detecting system based on an array of six QMB sensors to monitor the aroma profile changes during truffles storage. The variations (%) of the truffle volatile fingerprint registered by six e-nose sensors between the first and the 7th day of storage allow determining the best conditions of storage [51]. Zampioglou, and Kalomiros, (2014) proposed an EN method based on 6 low-cost MOS sensors array to differentiate truffles species (*Tuber borchii*, *Tuber macrosporum,* and *Tuber brumale*) according to their aroma fingerprints [80]. Since in truffle aroma studies, EN sensing is mainly used to characterize the overall aroma pattern of truffle samples, EN is usually coupled with other analytical techniques such as GC to identify, characterize, and quantify truffle aroma compounds.

## 4. Conclusions

The unique flavor of truffles is one of the main reasons to get worldwide attraction as a food product. Previous studies have focused on screening and identifying the volatile organic compounds (VOCs) and characterized more than 200 VOCs in various truffle species. The major compounds responsible for the aroma in diverse truffle species were well documented in our review which aimed to provide an update on research conducted on the analysis of active aroma components as well as the analytical techniques to identify them. Truffles possess significant variability in their aroma profiles from species to species. In general, sulfur compounds such as dimethyl sulfide (DMS) and dimethyl disulfide (DMDS), 1-octen-3-ol, and 2-methyl-1-propanol have been identified in most truffle species.

To deepen the knowledge about a complex odor of various truffle species, researchers have developed multiple methods to analyze truffle’s aroma. Traditionally, VOCs have been comprehensively profiled by solid-phase microextraction (SPME), which is generally followed by gas chromatography–mass spectrometry (GC–MS). The GC–MS based analysis expressed limitations to determine the correlation of quantified volatiles to the olfactory stimulus as this technique could not give information about human perception. Moreover, the perceived odor presented frequently at lower concentrations than the instrumental detection limit. To overcome these limitations and to identify the key aroma contributors amongst detected volatiles, the flavor dilution (FD) factor by aroma extract dilution analysis (AEDA) or/and the odor activity value (OAV) could be determined, which are based on the gas chromatography–olfactometry (GC–O) technology. In addition to these studies, numerous researchers performed GC–O based analysis together with SPME–GC–MS to find the critical aroma contributor among the comprehensively profiled truffle volatiles. The olfactometry based analysis considering the entire aroma mixture has been regarded as the most useful method for estimating the contribution of key aroma active compounds. In addition, the innovative technique time-of-flight (TOF–MS) based Proton Transfer Reaction-Mass Spectrometer (PTR-MS) technology was employed which improved the GC–MS based methods and provided a fast, accurate, and direct measurement of volatiles. In recent times, researchers have increasingly recognized the authenticity and traceability of flavor compounds in truffles. The authenticity and traceability can be determined by GC coupled with combustion-isotope ratio mass spectrometry (GC–C–IRMS), which exploits ^13^C/^12^C ratio abundance of the main aroma contributor in foods.

There are many areas need more attention. Greater attention is needed to discover how to benefit the knowledge of truffles’ aroma components and incorporate them into value-added truffle or truffles-related products. The quality of truffles can significantly vary from one species to another according to the aforementioned factors affecting truffle aroma and quality. So, standardization of truffles in terms of aroma profiles is very important and needs more investigation as there is a lack of R&D system in the identification and standardization of natural species of truffles. Furthermore, with the development of advanced analytical methods, researchers can identify the chemicals responsible for the truffles’ aroma profiles. The challenge is how to use these findings to enhance truffle functionality and how to employ these advanced analytic methods to on-line monitors during the processing of truffle-related products.

## Figures and Tables

**Figure 1 molecules-25-05948-f001:**
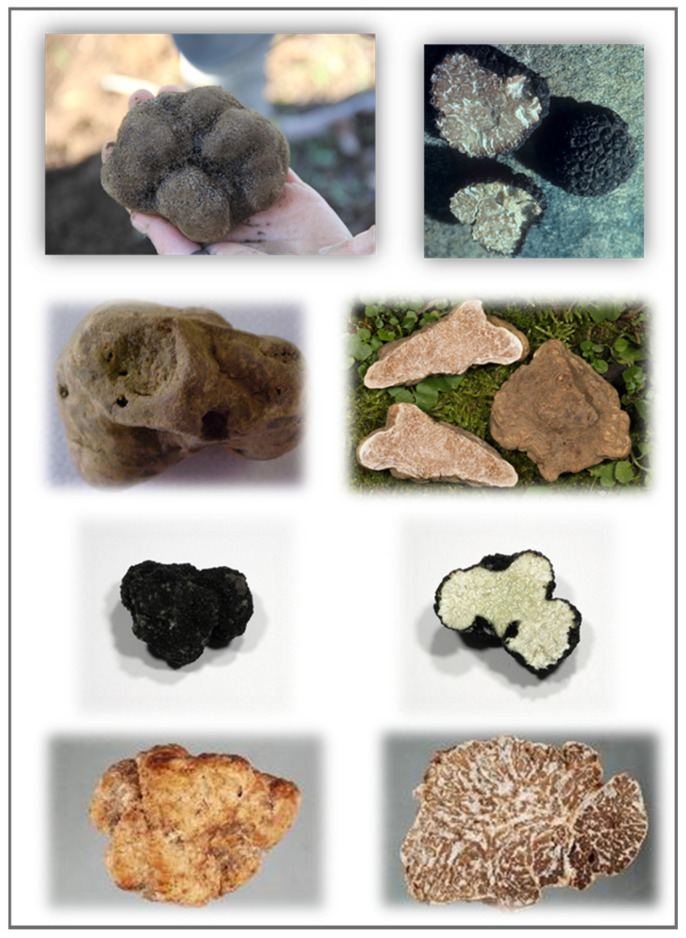
Most valuable truffles occur in Europe such as (from top to bottom) *T. melanosporum*, *T. magnatum*, *T. aestivum,* and *T. borchii*. (Photos have been provided by “Mycology Collections Portal” https://mycoportal.org/portal/collections/harvestparams.php?db%5B%5D=25&x=46&y=24, accessed on 20 November 2020).

**Table 2 molecules-25-05948-t002:** Major analytical methods used for the determination of aroma compounds in truffles.

	Sampling	Analyte Separation	Detecting System	Reference
Gas chromatography-based methods (GC).	1. Static Headspace (SHS). 2. Dynamic HS (DHS). 3. Purge and Trap (PT). 4. HS-Solid phase microextraction (HS-SPME). 5. Direct solvent extraction coupled with solvent-assisted flavor evaporation (DSE-SAFE).	1. Monodimensional GC: DB-WAX, HP-5MS. 2. Comprehensive two-dimensional GC (GC × GC).	1. Flame Ionization Detector (GC-FID)	[49,71]
			2. Mass spectrometry detector (GC-MS):	
			- Quadrupole MS (QMS)	[70]
			- Ion trap mass MS (IT-MS)	[72]
			- High-Resolution Time-of-Flight MS (HR-TOF).	[27]
			3. Olfactometry detector (GC-O)	[73,74,75]
Proton-transfer-reaction mass spectrometry (PTR-MS).	Direct injection of VOC from the HS.	Not applicable	1. High-Resolution Time-of-Flight MS (TOF). 2. QMS	[76,77,78]
Electronic nose (EN) sensing	SHS, DHS, PT, and HS-SPME	Not applicable	1. Metal-oxide sensors (MOS).	[79,80]
			2. Quartz microbalance (QMB) sensors.	[51,71]

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
