# Peer review of "An Overview on Truffle Aroma and Main Volatile Compounds"

_molecules, 2020, doi:10.3390/molecules25245948_

Round 1
Reviewer 1 Report
Dear all,
below are my few sugesstions or recomendations:
Manuscript ID: molecules-1032001
Title: An overview on truffle aroma and main volatile compounds
This review evaluate the aroma profiles of different species of truffles to deepen the knowledge about a complex odor of various truffles and an update of the analytical techniques to identify them.
Generally, it is a good written article.
L357 Pacioni et al., (2014) - delete comma
L424 Splivallo et al (2012) – add the point
L426 Costa et al. - , (2015) - delete comma
L 435 (NIST, WILEY, ADAMS, FFNSC…) – delete points
L448 Feng et al (2019) - add the point
L481 Pennazza 480 et al (2013) - add the point
I recommend that the Table 2 contains a column with references.
The number of bibliographic sources is adequate, more than 50% of the total bibliographic sources are from the last 5 years.
Please check your English (errors in spelling, grammar, and style)
Author Response
Reviewer#1
Title: An overview on truffle aroma and main volatile compounds
This review evaluate the aroma profiles of different species of truffles to deepen the knowledge about a complex odor of various truffles and an update of the analytical techniques to identify them.
Generally, it is a good written article.
L357 Pacioni et al., (2014) - delete comma
According to reviewer suggestion, the correction has been made.
L424 Splivallo et al (2012) – add the point
According to reviewer suggestion, the correction has been made.
L426 Costa et al. - , (2015) - delete comma
According to reviewer suggestion, the correction has been made.
L 435 (NIST, WILEY, ADAMS, FFNSC…) – delete points
According to reviewer suggestion, the correction has been made.
L448 Feng et al (2019) - add the point
According to reviewer suggestion, the correction has been made.
L481 Pennazza 480 et al (2013) - add the point
According to reviewer suggestion, the correction has been made.
I recommend that the Table 2 contains a column with references.
According to reviewer suggestion, we added a coulumn with references.
The number of bibliographic sources is adequate, more than 50% of the total bibliographic sources are from the last 5 years.
Thank you
Please check your English (errors in spelling, grammar, and style)
According to reviewer suggestion, the correction has been made.
Reviewer 2 Report
This review is interesting and the bibliography included is appropriate. However, I ask the authors to deepen the topic regarding the recognition of truffles from animals, such as wild boar and trained dogs. What are the molecules responsible for the identification? As a starting point, the authors could consider the work of Talou, T et al. 1990 Mycol Res. Letter to editor.
Author Response
Reviewer#2
This review is interesting and the bibliography included is appropriate. However, I ask the authors to deepen the topic regarding the recognition of truffles from animals, such as wild boar and trained dogs. What are the molecules responsible for the identification? As a starting point, the authors could consider the work of Talou, T et al. 1990 Mycol Res. Letter to editor.
According to reviewer suggestion, an additional paragraph (1.4) has been inserted to the manuscript and we considered the reference Talou, et al. 1990.
Round 2
Reviewer 2 Report
I thank the authors for answering my questions and including the bibliography